# Tunable High-Sensitivity Four-Frequency Refractive Index Sensor Based on Graphene Metamaterial

**DOI:** 10.3390/s24082658

**Published:** 2024-04-22

**Authors:** Xu Bao, Shujun Yu, Wenqiang Lu, Zhiqiang Hao, Zao Yi, Shubo Cheng, Bin Tang, Jianguo Zhang, Chaojun Tang, Yougen Yi

**Affiliations:** 1Joint Laboratory for Extreme Conditions Matter Properties, Key Laboratory of Manufacturing Process Testing Technology of Ministry of Education, State Key Laboratory of Environment-Friendly Energy Materials, Southwest University of Science and Technology, Mianyang 621010, China; baoxu0409@163.com (X.B.); 13615581367@163.com (W.L.); 2Key Laboratory of Metallurgical Equipment and Control Technology of the Ministry of Education, Wuhan University of Science and Technology, Wuhan 430074, China; yushujun@wust.edu.cn (S.Y.); haozhiqiang@wust.edu.cn (Z.H.); 3School of Physics and Optoelectronic Engineering, Yangtze University, Jingzhou 434023, China; shubocheng@yangtzeu.edu.cn; 4School of Chemistry and Chemical Engineering, Jishou University, Jishou 416000, China; 5School of Microelectronics and Control Engineering, Changzhou University, Changzhou 213164, China; btang@cczu.edu.cn; 6Department of Physics, Jinzhong University, Jinzhong 030619, China; phys.zhangjg@gmail.com; 7College of Science, Zhejiang University of Technology, Hangzhou 310023, China; chaojuntang@126.com; 8College of Physics and Electronics, Central South University, Changsha 410083, China; yougenyi@csu.edu.cn

**Keywords:** graphene, high sensitivity, multi-frequency sensor, tunability, polarization insensitivity

## Abstract

As graphene-related technology advances, the benefits of graphene metamaterials become more apparent. In this study, a surface-isolated exciton-based absorber is built by running relevant simulations on graphene, which can achieve more than 98% perfect absorption at multiple frequencies in the MWIR (MediumWavelength Infra-Red (MWIR) band as compared to the typical absorber. The absorber consists of three layers: the bottom layer is gold, the middle layer is dielectric, and the top layer is patterned with graphene. Tunability was achieved by electrically altering graphene’s Fermi energy, hence the position of the absorption peak. The influence of graphene’s relaxation time on the sensor is discussed. Due to the symmetry of its structure, different angles of light source incidence have little effect on the absorption rate, leading to polarization insensitivity, especially for TE waves, and this absorber has polarization insensitivity at ultra-wide-angle degrees. The sensor is characterized by its tunability, polarisation insensitivity, and high sensitivity, with a sensitivity of up to 21.60 THz/refractive index unit (RIU). This paper demonstrates the feasibility of the multi-frequency sensor and provides a theoretical basis for the realization of the multi-frequency sensor. This makes it possible to apply it to high-sensitivity sensors.

## 1. Introduction

The optical absorber plays a critical role in sensing [1,2,3,4], imaging [5,6], solar energy consumption [7], and other applications. Conventional absorbers consist mainly of a top metal layer, an intermediate dielectric layer, and a bottom metal layer, which use the metal’s surface with equipartitioned excitations to confine the electromagnetic wave and achieve a high absorption peak [8,9]. Because of the advancement of technology, the performance of narrow-band absorbers has increased; typical narrow-band absorbers are unable to satisfy the demand, so the most promising material for an absorber is a graphene metamaterial absorber. A tunable high-sensitivity absorber can be realized by taking advantage of the specific features of graphene metamaterial.

A layer of C atoms form covalent bonds with one another through sp^2^ hybridization to generate graphene, a hexagonal honeycomb planar structure. The primary cause of graphene’s planar conductivity is the electrons that each C atom possesses that are not involved in bonding [10,11,12]. Monolayer graphene has a unique energy band structure that results in linear energy dispersion and two crossed cones at the Dirac point, meaning that there is no bandgap. As a result, it has an ultra-high carrier mobility rate, a high specific surface area, high thermal conductivity, and unique optoelectronic properties [13,14]. However, the low absorptivity of monolithic homogenous graphene (approximately 2.3% absorption) makes it difficult to transform into an ideal absorber. In 2008, Landy et al. developed the idea of a graphene perfect absorber [15]. They discovered that surface plasmon resonance occurs in graphene when it is used as an ultra-surface layer in an absorber. Tunable graphene Fermi energy level allows for efficient terahertz wave reflection, absorption, and transmission [16,17]. Graphene is an excited element with excellent confinement, low loss, and dynamic tunability. More crucially, by adjusting the voltage delivered to graphene via an external gate, the graphene absorber can obtain tunable characteristics suitable for a variety of applications [18,19,20]. This prompted later researchers to develop a variety of graphene absorbers, including narrow-band [21,22], broadband [23,24], and multi-frequency absorbers [25,26,27].

This work proposes an adjustable and highly sensitive quad-frequency refractive index sensor based on graphene metamaterials. The absorber has graphene on top, a dielectric layer in the middle, and a gold layer on the bottom. The absorber features four absorption peaks, three of which are perfect, with absorptivities of 0.98965, 0.99857, 0.96241, and 0.99108, respectively. The absorptivities of the absorber are varied by adjusting the Fermi energy level of graphene. Adjusting the Fermi energy level of graphene changes the absorber’s absorptivities. Furthermore, the absorber performance was improved by adjusting the graphene relaxation time and structural characteristics using modeling tools. A correlation study on the absorber’s sensitivity was conducted, and the conclusion was that the absorber had a high sensitivity. The maximum sensitivity of the four absorption peaks is 21.60 THz/RIU, providing it with an exceedingly rare advantage over other absorbers. Overall, the absorber has numerous applications in high-sensitivity sensors [28,29].

## 2. Introduction to the Theory and Structural Model

In this article, a tunable multi-frequency absorber is obtained through modeling. Its structure is shown in Figure 1a, where the thickness of the gold layer is T_1_ = 540 nm. The dielectric layer in the middle of this absorber has a refractive index of 1.60 and the thickness of this layer is T_2_ = 600 nm. Its basic unit is shown in Figure 1b, and the geometrical parameters of the structure are P = 700 nm, L_1_ = 100 nm, L_2_ = L_3_ = 50 nm, L_4_ = 30 nm, and R = 140 nm.

The gold used in this paper is the classical Drude model of lossy gold, which has the following dielectric constant [30]:(1)$$\varepsilon_{Au}=\varepsilon_{\infty }-\frac{\omega_{p}^{2}}{\omega_{p}^{2}+i\omega \gamma }\;$$
where $\varepsilon_{\infty }\;$= 9.1 is the high-frequency limit dielectric constant. The plasmon frequency $\omega_{p}\;$ = 1.3659 × 10^16^ rad/s and the damping frequency γ = 1.0318 × 10^14^ rad/s are used.

The skinning depth of electromagnetic waves in a good conductor decreases as the frequency, electrical conductivity, and magnetic permeability of the medium increase. The maximal skinning depth for gold in the THz band is 266 nm. The thickness of the gold layer we utilized is substantially more than the skinning depth of the electromagnetic waves; hence, the layer entirely prevents electromagnetic wave propagation [31].

The intermediate dielectric layer is made of silicon oxide. In the MWIR band, SiO_2_ has a refractive index of 1.3 to 1.4. The refractive index of SiO_2_ in this study is 1.6. TiO_2_ is doped into SiO_2_ via plasma-enhanced chemical vapor deposition [32], altering the crystalline structure of SiO_2_ and, hence, its refractive index. The Si/Ti ratio was set to ensure that the dielectric layer’s refractive index was 1.6.

The top layer is patterned with graphene in the shape of an open mushroom plus four little squares, as shown in Figure 1b. The dielectric constant of graphene in a single layer can be stated as follows [33]:(2)$$\varepsilon \left(\omega \right)=1+\frac{i\delta_{g}}{\omega \varepsilon_{0}\Delta }\;$$
where $\delta_{g}$ is the conductivity of graphene, ω is the radian frequency, $\varepsilon_{0}$ is the vacuum dielectric constant, Δ is the thickness of graphene, and the thickness of graphene is 1 nm.

The conductivity of graphene is divided into two components, intraband conductivity $\sigma_{intra}$ and interband conductivity $\sigma_{inter}$, which can be expressed via Kubo equation as follows [34,35]:(3)$$\sigma \left(\omega ,\Gamma ,\mu_{c},Τ\right)=\sigma_{intra}\left(\omega ,\Gamma ,\mu_{c},Τ\right)+\sigma_{inter}\left(\omega ,\Gamma ,\mu_{c},Τ\right)$$
(4)$$\sigma_{intra}\left(\omega ,\Gamma ,\mu_{c},Τ\right)=\frac{ie\left(\omega +i2\Gamma \right)}{\pi \hbar^{2}}\left[\frac{1}{{\left(\omega +2i\Gamma \right)}^{2}}\int_{0}^{\infty }\varepsilon \left(\frac{\partial f_{d}\left(\varepsilon \right)}{\partial \varepsilon }-\frac{\partial f_{d}\left(-\varepsilon \right)}{\partial \varepsilon }\right)d\varepsilon \right]$$
(5)$$\sigma_{inter}\left(\omega ,\Gamma ,\mu_{c},Τ\right)=\frac{ie\left(\omega -i2\Gamma \right)}{\pi \hbar^{2}}\left[\int_{0}^{\infty }\frac{f_{d}\left(-\varepsilon \right)-f_{d}\left(\varepsilon \right)}{{\left(\omega -2i\Gamma \right)}^{2}-4{\left(\varepsilon /\hbar \right)}^{2}}d\varepsilon \right]$$

In the formula, ω is the angular frequency, e is the elementary charge, Γ is the scattering rate Γ=1/2τ, τ is the relaxation time, $\mu_{\;c}$ is the chemical potential, ℏ is Planck’s constant, and $k_{B}$ is Boltzmann’s constant. $f_{d}\left(\varepsilon \right)$ is the Fermi Dirac distribution [36,37]:(6)$$f_{d}\left(\varepsilon \right)={\left[exp\left(\varepsilon -\mu_{c}/kΤ\right)+1\right]}^{-1}$$

At low light frequencies, graphene has zero interband conductivity. As a result, only intraband conductivity is included when calculating graphene conductivity. Specifically, this can be stated as follows [38]:(7)$$\sigma \left(\omega \right)=\frac{ie^{2}\mu_{c}}{\pi \hbar^{2}\left(\omega +i2\Gamma \right)}$$

This model is computationally simulated using the FDTD module in the Ansys Lumerical software (Ansys Lumerical 2023), in which the relevant graphene parameters in the FDTD material library are varied so that the Fermi energy level of graphene is 0.94 eV and the relaxation time is 1 ps. The periodic boundary conditions are set in the x and y directions, with the z direction set to be a perfectly matched layer. The model in FDTD is set up as follows: the x and y directions are set up as periodic boundary conditions, and the z direction is set up as a perfectly matched layer (PML); the mesh is set up as a 40 nm step in the x and y directions and 1 nm in the z direction in the plane where the graphene material is located. The model assumes that the ambient refractive index is 1. In this paper, the incident wave is a plane wave that occurs above the graphene layer. The incident wave has equal intensities at each wavelength and is classified as a TE wave. The absorber absorptivity is determined via the following formula [39,40,41]:(8)A=1−R−T
where A is absorptivity, R is reflectivity, and T is transmittance. Because the bottom layer is gold, and its thickness is significantly more than its skinning depth, no light can flow through the gold layer; thus, set T = 0. This absorber absorptivity formula is reduced to the following:(9)A=1−R

When R ≤ 0.02, the absorption peak is recognized as perfect absorption.

However, due to the intricacy of the absorber design and material selection, developing a unified analytical model for the absorber’s resonant absorption is extremely challenging. However, the fundamental process of its electrodynamics is evident and can be explained in the following manner:

Driven by an applied electromagnetic field (i.e., incident light), free electron collective oscillations pair with the external electromagnetic field, resulting in a localized increase in the electromagnetic field. The localized enhancement of the electromagnetic field increases the dipole moments of the molecules on the surface of the nanomaterial, and when the frequency of the surface-parity excitation coincides with the oscillation frequency of the molecules, the two produce a coupling, which greatly improves the material’s absorption. As a result, the field amplification effect can be achieved by modifying the structural gap between adjacent units in the equipartitioned exciton metamaterial.

## 3. Results and Discussion

As shown in Figure 2a, the order of magnitude of the transmittance is around 10^−9^, which is absolutely inconsequential because the thickness of the gold layer is greater than the skin-convergence depth of the electromagnetic wave in the gold element, as previously mentioned. The absorber exhibits five distinct absorption peaks: models Ⅰ, II, III, IV, and V. The absorption peaks’ resonance frequencies are 69.0449 THz, 69.4769 THz, 71.6372 THz, 73.8215 THz, and 74.7096 THz, respectively, with absorptivities of 0.98572, 0.99787, 0.96395, 0.98922, and 0.34485. In Figure 2a, it is clear that model 5’s absorption rate is extremely low and of little practical consequence, and it will not be discussed further.

TE waves are transverse electric waves, which are electromagnetic waves in which the electric field vibrates perpendicular to the direction of wave propagation while the magnetic field vibrates parallel to the direction of wave propagation. TM waves are transverse magnetic waves, which are electromagnetic waves with the magnetic field vibrating perpendicular to the direction of wave propagation and the electric field vibrating parallel to the direction of wave propagation. This is an advantage that the non-centrosymmetric model does not have. The absorption spectra in the TE-wave and TM-wave modes are identical, as shown in Figure 2b, owing to the model’s core symmetry [42,43]. This is one advantage that the non-centrosymmetric model does not have.

Figure 3 shows the distribution of the absorber’s electric field intensity at the four absorption peak frequencies. At absorption peak Ⅰ, the electric field strength is localized in the region near the left and right holes. This is due to the electric dipole resonance of the surface plasma of the single-layer graphene coupled with the electromagnetic field. This resonance locally enhances the electromagnetic field, making the incident wave much more absorptive [44]. At absorption peak Ⅱ, the electric field strength is localized in the region near the upper and lower holes. At absorption peak Ⅲ, the electric field strength is localized in the region near the upper and lower semicircle of the upper and lower holes, as well as the region near the right angle formed by the large square. At absorption peak Ⅳ, the electric field strength is localized in the area near the small square. As in the preceding statement, the interaction between the collective oscillation of free electrons and the electromagnetic field causes the electric field to become highly localized, allowing the absorber to perfectly absorb light of a given frequency.

As shown in Figure 4, the absorber’s polarization insensitivity is essential in practical applications since the absorber will not have ideal optical conditions in real life. The figure shows that, for the absorber under the TE wave, when the incident light is between 0° and 90°, the absorption peak absorbance in the absorber’s absorption spectrum does not change much. At the TM wave, when the incident light is between 0° and 45°, the absorber’s absorption peak absorbance does not change significantly. Our absorber thus exhibits ultra-wide-angle polarization insensitivity for TE waves and high-angle polarization insensitivity for TM waves. This demonstrates that the absorber has a wide range of practical applications in both production and life, as well as promising future applications [45,46,47].

In the previous section, we investigated the effect of the absorber’s electric field distribution at the resonance frequency of the absorption peak, as well as the angle of incidence of the external optical mode, on the absorber. The impact of the absorber’s intrinsic qualities on the absorber and its mode of action is also investigated. The Fermi energy levels and relaxation times of graphene, as well as the dielectric layer’s refractive index, are modified, and the impacts on the absorption spectra are investigated. The Fermi energy levels of graphene can be dynamically adjusted by applying an external DC bias voltage $V_{g}$ to the graphene layer through an ion gel. The link between them is as follows [48]:(10)$$E_{F}=V_{F}\sqrt{\pi \varepsilon_{0}\varepsilon_{r}V_{g}/e_{0}t_{s}}$$
where $V_{g}$ is the external voltage, e_0_ is the electron charge, V_F_ is the Fermi velocity, and t_s_ is the thickness of the dielectric layer.

As illustrated in Figure 5a, the position of this absorber’s absorption peak varies as the Fermi energy level of the graphene changes. The absorber’s absorption peak shifts blue as the Fermi energy level of the graphene increases. This effect could be attributed to the more difficult excitation of nonequilibrium carriers caused by an increase in the Fermi energy level, necessitating the use of higher-frequency electromagnetic waves to excite the creation of nonequilibrium carriers in graphene [49,50]. These phenomena have a significant impact on the tunability of the absorber. It is worth noting that changing the graphene Fermi energy level has only a minor effect on the absorbance of the absorption peaks, indicating that the absorber performs well. The graphene Fermi energy level rises from 0.90 eV to 0.98 eV, and the resonance frequency of the absorption peaks shifts from 66.1324 THz to 69.0363 THz, 66.5657 THz to 69.4813 THz, 68.6265 THz to 71.6358 THz, and 70.7108 THz to 73.8137 THz. As a result, graphene’s Fermi energy level is electrically tuned, making the absorber extremely tunable, which is one of its benefits as a metamaterial absorber.

The effect of graphene’s relaxation time on the absorption spectrum is explored below, beginning with the relaxation time (an important graphene property), using this expression [51,52]:(11)$$\tau =\mu \nu /eV_{F}^{2}$$
where μ and ν are the chemical potential and carrier mobility of graphene, respectively, e represents the charge of the meta-charge, and V_F_ is the Fermi velocity of graphene, which can be calculated as $V_{F}=10^{6}\;m/s$, according to the simple energy band theory.

Figure 5b depicts the fluctuations in this absorber’s absorption spectrum with graphene relaxation time. The figure shows that the relaxation time of graphene has little effect on the resonance wavelength of the absorber’s absorption peaks, whereas it can affect the absorbance of the absorption peaks. When the relaxation time is 0.2 ps, the first two peaks are united to form a single peak with a resonance wavelength of 69.3701 THz, which is placed between 69.0363 and 69.4813 THz, while the first two peaks gradually separate as the relaxation time increases. The relaxation time of graphene has a significant impact on the absorbance of absorption peaks within a specific range (0.4 ps to 1.0 ps, 0.81325 to 0.99108, 0.88137 to 0.99732, 0.94208 to 0.95948, and 0.76604 to 0.99237). This occurrence shows that graphene’s relaxation time can better manage the absorbance of the absorption peaks, resulting in the effect of perfect absorption [53,54]. Because graphene’s relaxation time is immutable at the moment of production, this paper uses graphene with a relaxation time of 1 ps.

Figure 6 depicts the effect of the dielectric layer’s refractive index, SiO_2_, on the absorber’s absorption spectrum. Figure 6b shows that the effect of the dielectric layer’s refractive index on the absorbance of the absorption peaks is irregular. The absorption rate varies very little; absorption rate of absorption peak I varies from 0.98766 to 0.98865 with a fluctuation of 0.099%; the absorption rate of absorption peak II varies from 0.99698 to 0.99947 with a fluctuation of 0.249%; the absorption rate of absorption peak III varies from 0.93871 to 0.96263 with a fluctuation of 2.392%; and the absorption rate of absorption peak IV varies from 0.98831 to 0.99088 with a fluctuation of 0.257%. These results show that the dielectric layer’s refractive index has very little effect on the absorber’s absorption rate. Figure 6c indicates that the absorption peak’s resonance frequency has a linear connection with the dielectric layer’s refractive index. This was determined by analyzing the data and fitting the refractive index n and resonant frequency γ with a linear function:(12)$$\gamma_{Ⅰ}=-26.634n+111.6634$$
(13)$$\gamma_{Ⅱ}=-26.126n+111.3081$$
(14)$$\gamma_{Ⅲ}=-26.126n+111.3081$$
(15)$$\gamma_{Ⅳ}=-28.552n+119.5233$$

Equations (12)–(15) apply to dielectric layers with a refractive index of 1.6–1.9. As previously stated, the resonance frequency of the absorber can be tuned by adjusting the Fermi energy level of graphene; however, the precise tuning of the absorber is difficult due to the complex functional relationship between the Fermi energy level of graphene and the resonance frequency. The linear relationship between the refractive index of the dielectric layer and the resonance frequency described above, which has a weak effect on the absorptivity, allows for accurate tuning of the absorber [55,56]. Using the equation above, the absorber may detect changes in the refractive index of the dielectric layer material.

By changing the parameters of the Fermi energy level and the relaxation time of graphene, as well as the refractive index of the dielectric layer, to understand the changes in the absorption spectrum of the absorber, we can conclude the following: the Fermi energy level of graphene is able to change the resonance wavelength of the absorption peak; the relaxation time of graphene is able to affect the absorption rate of the absorption peak of the absorber; the refractive index of the dielectric layer mainly affects the resonance wavelength of the absorption peak of the absorber. Moreover, the resonance wavelength and the refractive index show linearity, and have very little effect on the absorbance rate.

Finally, we analyze one of the most important parameters of this absorber, the sensitivity(S) and figure of merit (FOM) of the absorber.

A sshown in Figure 7, absorbers play a critical role and are used extensively in sensing and detection. An absorber can be utilized with the ultra-sensitive atmospheric refractive index sensor. As shown in Figure 7b, the absorbance of the absorption peak varies in a complex manner with the air refractive index, which has little practical relevance in the actual application. However, Figure 7c shows that its absorption rate has a linear connection with the ambient refractive index, which is a remarkable phenomenon. Using this approach, we can determine the change in the atmospheric refractive index by changing the resonance frequency of the absorption peak, demonstrating the effect of atmospheric refractive index detection [57,58]. We shall evaluate this absorber’s performance in terms of sensitivity (S) and quality factor (FOM). The formulas for these two metrics are as follows [59,60,61]:(16)$$S=\frac{\Delta \lambda }{\Delta n}$$
(17)$$FOM=\frac{S}{FWHM}$$
where Δλ is the amount of change in the resonance wavelength of the absorption peak with respect to the change in the ambient refractive index. FWHM is the half-height width of the absorption peak. Using a simple substitution calculation, the S of these four peaks can be obtained as follows: 19.5025 THz/RIU, 21.6025 THz/RIU, 20.7025 THz/RIU, and 20.7025 THz/RIU. The sensitivity of this sensor is an extremely large advantage compared to other absorbers, so the use of this absorber in this refractive index sensor has great potential. This absorber has great potential for refractive index sensors [62,63,64].

The FOM is a useful measure of the absorber’s bandwidth. To compute the FWHM of all peaks, the background refractive index is set to Nc = 1. The FWHM of absorber peak Ⅰ and absorber peak Ⅱ cannot be determined due to their proximity. The FOM of this absorber’s last two peaks can be calculated as 49.06 and 71.66, respectively. Furthermore, we compared our absorber-to-absorber research from recent years and, as shown in Table 1 [65,66,67,68,69], our absorber has high sensitivity and polarization insensitivity, as well as a significant number of absorption peaks, demonstrating the strength of our absorber.

## 4. Conclusions

In this paper, we designed a tunable four-frequency absorber MWIR, where the top layer of the absorber is patterned with graphene, which consists of a dug-out mushroom shape and four small squares. It has four absorption peaks with three perfect absorption peaks. We explained the absorption mechanism of this absorber using electric field theory. In addition, we adjusted the Fermi energy level and relaxation time of graphene, obtained the tuning of the absorber by adjusting the refractive index of the intermediate dielectric layer, and verified that the resonance frequency of the absorption peaks and the absorption rate of the absorber had a very good tuning ability. The practicality of the absorber was also analyzed; it is characterized by a high sensitivity compared to other absorbers, with the highest sensitivity of 21.60 THz/RIU. In addition, the absorber achieves polarization insensitivity at an incident angle of 0° to 90° for TE waves and achieves polarization insensitivity at an incident angle of 0° to 45° for TM waves, and the absorber achieves insensitivity with an ultra-wide angle for TE waves compared to other absorbers. We believe that our absorbers will be used in a wide range of applications in the field of high-sensitivity detection and optoelectronic detection.

## Figures and Tables

**Figure 1 sensors-24-02658-f001:**
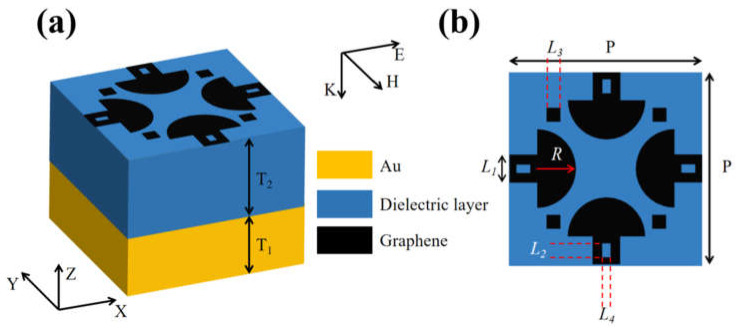
(**a**) Basic unit of graphene absorber; (**b**) top view of absorber.

**Figure 2 sensors-24-02658-f002:**
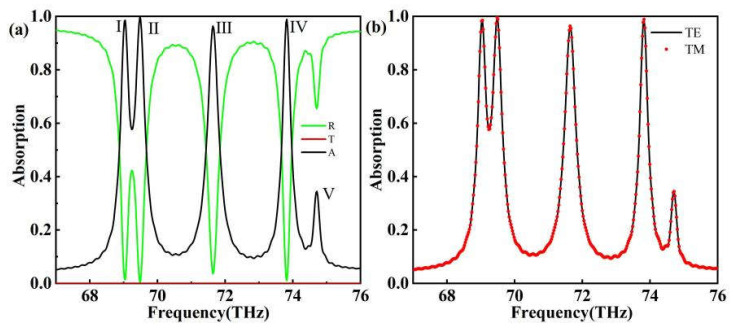
(**a**) Reflection/transmission/absorption spectra of absorber; (**b**) absorption spectra of TE and TM waves. The black solid line is the absorption spectrum of the TE wave and the red dashed line is the absorption spectrum of the TM wave.

**Figure 3 sensors-24-02658-f003:**
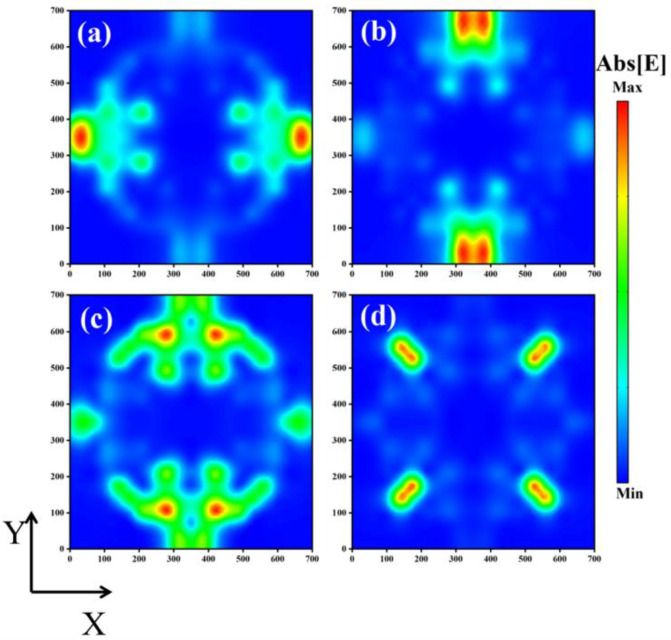
(**a**–**d**) show the electric field strength distribution on the absorber top (i.e., the graphene pattern) at the absorption frequencies of 69.0363 THz, 69.4813 THz, 71.6358 THz, and 73.8137 THz, respectively.

**Figure 4 sensors-24-02658-f004:**
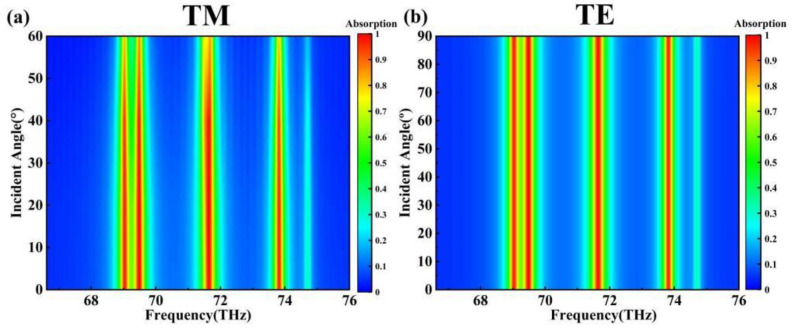
(**a**) TM wave angle scan; (**b**) TE wave angle scan.

**Figure 5 sensors-24-02658-f005:**
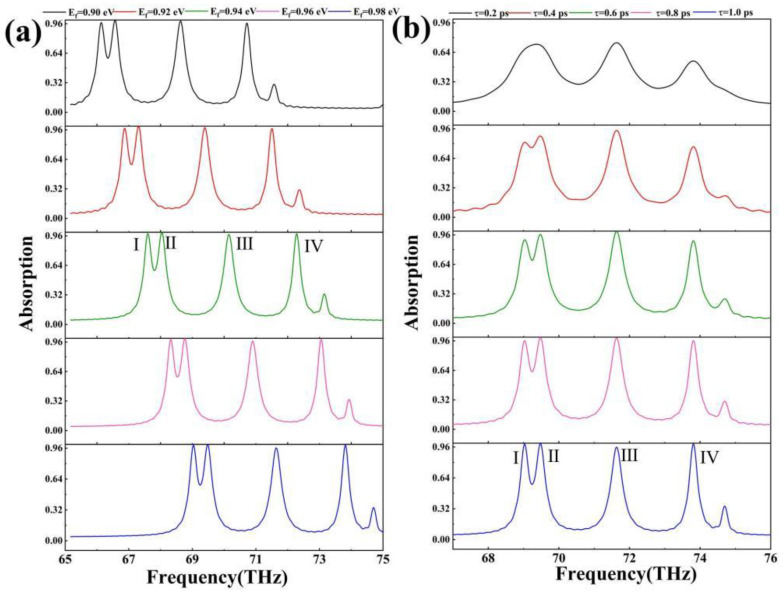
(**a**) Absorption spectra of graphene with Fermi energy levels of 0.90 eV, 0.92 eV, 0.94 eV, 0.96 eV, and 0.98 eV for the absorber; (**b**) absorption spectra of graphene with relaxation times of 0.2 ps, 0.4 ps, 0.6 ps, 0.8 ps, and 1.0 ps for the absorber.

**Figure 6 sensors-24-02658-f006:**
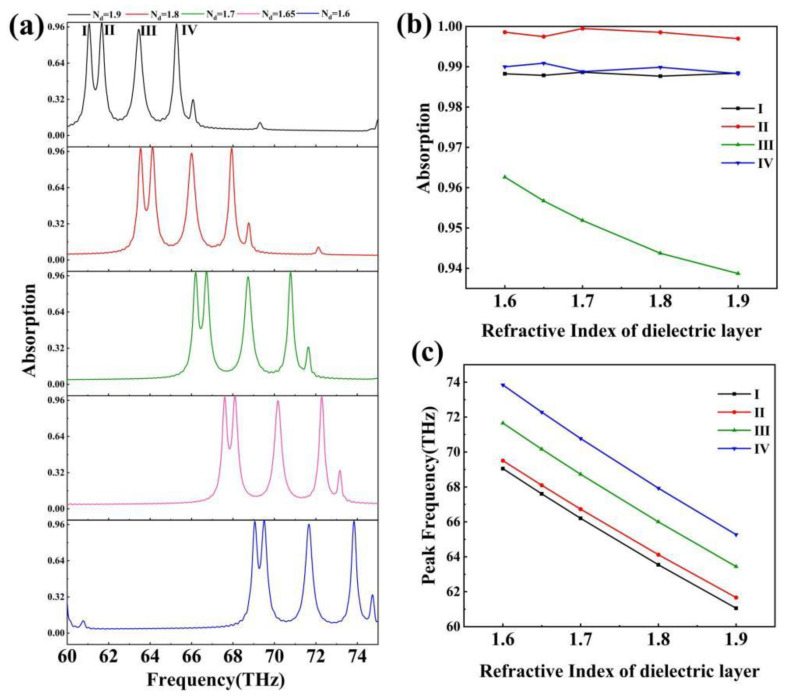
(**a**) Change in the absorption spectra of the absorber as the refractive index of the dielectric layer increases from 1.60 to 1.90; (**b**) changes in the absorbance of the four absorption peaks as the refractive index of the dielectric layer changes; (**c**) changes in the resonance frequency of the four absorption peaks as the refractive index of the dielectric layer changes.

**Figure 7 sensors-24-02658-f007:**
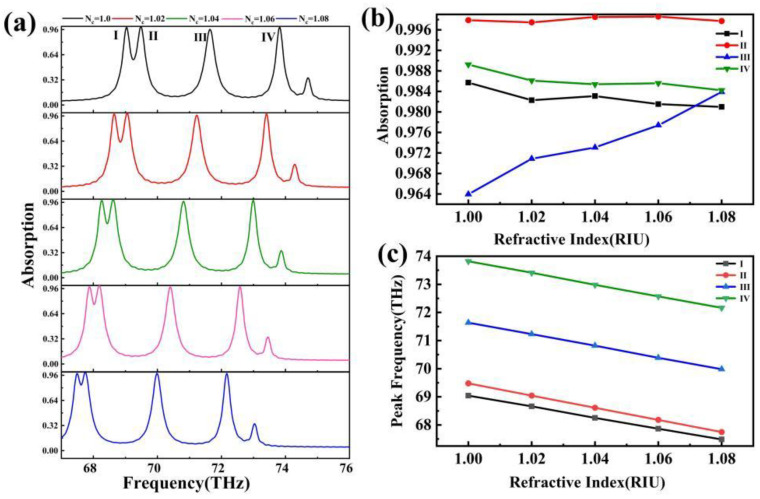
(**a**) Change in the absorption spectra of the absorber as the ambient refractive index increases from 1.00 to 1.08; (**b**) changes in the absorbance of the four absorption peaks with the change in the refractive index of the ambient refractive index; (**c**) changes in the resonance frequency of the four absorption peaks with the change in the refractive index of the ambient refractive index.

**Table 1 sensors-24-02658-t001:** Comparison with previous absorbers.

Reference	Peak Number	Average Absorption	Polarization Insensitive	Incident Angle	Sensitivity (THz/RIU)	FOM
[65]	2	99.0%	yes	0–60	2.475	76.89
[66]	2	99.9%	-	0–80	4.72	13.88
[67]	1	99.96%	yes	0–75	3.923	6.11
[68]	2	96.05%	-	0–50	1.84	-
[69]	2	99.00%	yes	-	14.88	53.09
This work	4	98.54%	yes	0–45	20.62	60.36

## Data Availability

Publicly available datasets were analyzed in this study. These data can be found here: https://www.lumerical.com/ (accessed on 1 January 2020).

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
