# Peer review of "Tunable High-Sensitivity Four-Frequency Refractive Index Sensor Based on Graphene Metamaterial"

_sensors, 2024, doi:10.3390/s24082658_

Round 1
Reviewer 1 Report
Comments and Suggestions for Authors
The paper aims at simulation of MWIR metamaterial absorber (not THz as written it the title) based on patterned graphene. The absorber has multiple layers – layer with graphene, dielectric layer and gold layer. Such layered structures are not new at all. There are a lot of works investigating the particular combination with the variation of the patterns.
The work is not free of shortcomings.
1. The title of the paper must be changed because MWIR range is considered in the text but not the THz one. The same change must be done for Abstract where the authors wrote about FIR range.
2. The language is bad. There are grammar mistakes, misprints, and repetitions of phrases (for instance, p2 in the middle of the first paragraph, text between p4 and p5, etc.), some sentences are too long; there are a lot of noun stacks. The text is carelessly written. It is rather difficult to understand the paper in the current state.
3. There are abbreviations in the text without explanation: MPA, RIU, TM, TE. Even if the abbreviation is often used in the periodic it must be explained.
4. P2, second paragraph “c atom” -> “C atom”
5. P2, third paragraph “in the order of their absorptivities” What do the authors mean by this phrase?
6. P2, third paragraph and all over the text “a good linear relationship” – "relationship" is a bad way to describe linear ratio or linear dependence.
7. Eq. 1 is ill written, badly described, and needs reference. What for is “100%” placed in Eq. 1. What is “9” in it? Where is the description of all variables of this equation? Actually all equations and special symbols need some correction all over the text.
8. P3, under Fig 1. There are not link to the Fig. 1 in the text. The authors use “Fig. 2” instead of “Fig. 1” in the text.
9. What was the reason for choosing the particular geometry? Why were the particular dimensions of the pattern used? Were these dimensions optimized and if not why?
10. P3, second paragraph. The paragraph must be rewritten as well as Eqs. 2 -4.
11. Eq. 2 is trivial. Why do the authors show it?
12. Variables in Eqs. 2-3 are not introduced.
13. Eqs. 5, 9, 14 need reference.
14. Check the paragraph under Eq. 5.
15. For all formulas shown, application limits must be specified.
16. The authors claim that they use some dielectric with n=1.6. However, on p8 SiO2 is mentioned as a dielectric. SiO2 has n=1.3 – 1.4 in the MWIR range. In this case, if the authors use n=1.6 for SiO2 all calculations are wrong.
17. All over the text, “ambient” – Do the authors mean “background”?
18. P4, paragraph after Eq. 10. “The first term … Eq. …” Term of which Eq. do the authors mean?
19. FDTD is the method and not a program. Do authors use CST, LUMERICAL, COMSOL or their own program? In the latter case there must be references to this program’s validity or the simulations should be compared with some experimental or good theoretical data. If the authors use some well-known program, its name also must be mentioned in the text and cited. Without it the paper cannot be published.
20. Was it a unit cell simulation? Such a structure should also be considered as an array as interaction between metamaterial cells (coupling between cells) can also affect the absorption.
21. There is no information about incident wave parameters in the text. This information must be added.
22. P4, three paragraphs before Section 3. These paragraphs must be rewritten to be clear.
23. If authors numbered absorption peaks on p5, these numbers must be added to the Fig. 2 and its caption.
24. Check the paragraph below Fig. 2.
25. Fig. 3 must be improved. There is not a palette axis label and field type. Is it Abs[E], Real [Ex] or something else?
26. P5, paragraph after Fig 3. What do the authors mean by the “field intensity of the absorber”? Is it an intensity of some component of E field on the absorber surface? The paragraph must be rewritten to be clear.
27. Do the authors define x-polarized light as the TE mode and y-polarized light as the TM mode?
28. Figs.5a,6a,7a must be enlarged.
29. How do authors plan to apply voltage to isolated square parts of the graphene pattern?
30. How to change the relaxation time without changing the structure or substrate?
31. There are two designations of the Fermi velocity in the text: Vf and vf.
33. What for do the authors show Eqs. 16 – 19?
33. P8, paragraph below Eq. 19. “which is a common method used in the previous study” Which study?
34. P8. Check letter size for SiO2.
35. P9, paragraph after Eq. 21. “FOM” -> “S”
36. It is not clear how was FWHM calculated for the first and second peaks? For which value of the background refractive index was FWHM calculated for all peaks?
37. Conclusion. “”Far-infrared” -> “MWIR”
The paper is ill structured. It seems that the authors tried to separate Result and Discussion (see page 6 on top and page 8 after Eq. 19) into several subsections but failed with it.
So, the paper needs improvements before it can be considered for publication again.
Besides, this paper looks like yet another work about THz absorber simulations without any validation. The novelty is not evident.
So I do not recommend it for publication in the current state.
Author Response
Dear editor and reviewer:
We are grateful to the editor and reviewers for their constructive comments and suggestions on the revision of the manuscript. We have made all the necessary changes as suggested by the editor and reviewers. All revisions and supporting information in the manuscript are highlighted in red.
Reviewer #1:
Comment 1:The title of the paper must be changed because MWIR range is considered in the text but not the THz one. The same change must be done for Abstract where the authors wrote about FIR range.
Response:Thanks for your valuable counsel.According to your suggestion, we give the answer following.
We changed the title to "Tunable high-sensitivity four-frequency refractive index sensor based on graphene metamaterial"
Comment 2:The language is bad. There are grammar mistakes, misprints, and repetitions of phrases (for instance, p2 in the middle of the first paragraph, text between p4 and p5, etc.), some sentences are too long; there are a lot of noun stacks. The text is carelessly written. It is rather difficult to understand the paper in the current state.
Response:Thanks for your valuable counsel.According to your suggestion, we give the answer following. We revised the article extensively, checking it for grammatical errors, optimizing the language and enhancing its readability.
Comment 3:There are abbreviations in the text without explanation: MPA, RIU, TM, TE. Even if the abbreviation is often used in the periodic it must be explained.
Response:Thanks for your valuable counsel.According to your suggestion, we give the answer following. In our text, we have added to these abbreviated words, we have added to them. In the text for:
- which can achieve more than 98% perfect absorption at multiple frequencies in the MWIR(Medium Wavelength Infra-red) band compared with the traditional absorber.
- polarisation insensitivity and high sensitivity, with a sensitivity of up to 21.60 THz/RIU (Refractive index unit).
- TE waves are transverse electric waves, which are electromagnetic waves in which the direction of vibration of the electric field is perpendicular to the direction of wave propagation, while the direction of vibration of the magnetic field is along the direction of wave propagation. TM waves are transverse magnetic waves, which are electromagnetic waves in which the direction of vibration of the magnetic field is perpendicular to the direction of wave propagation, while the direction of vibration of the electric field is along the direction of wave propagation.
Comment 4:P2, second paragraph “c atom” -> “C atom”
Response:Thanks for your valuable counsel.According to your suggestion, we give the answer following. We checked and corrected the letter case problem in the text. In the text for:
Graphene is a hexagonal honeycomb planar structure formed by covalent bonding of a layer of C atoms through sp2 hybridization. Each C atom in graphene has an electron that is not involved in bonding,
Comment 5:P2, third paragraph “in the order of their absorptivities” What do the authors mean by this phrase?
Response:Thanks for your valuable counsel.According to your suggestion, we give the answer following. Our language errors, and what we want to express, are in order. In the text for:
and the absorptivities according to the four peaks are 0.98965, 0.99857, 0.96241, and 0.99108, respectively, in order.
Comment 6:P2, third paragraph and all over the text “a good linear relationship” – "relationship" is a bad way to describe linear ratio or linear dependence.
Response:Thanks for your valuable counsel.According to your suggestion, we give the answer following. After considering the article as a whole, we found the narrative to be of little practical significance and the narrative has been deleted.
Comment 7:Eq. 1 is ill written, badly described, and needs reference. What for is “100%” placed in Eq. 1. What is “9” in it? Where is the description of all variables of this equation? Actually all equations and special symbols need some correction all over the text.
Response:Thanks for your valuable counsel.According to your suggestion, we give the answer following. We modified Equation 1, referring to the article on the subject. The "9" was an accidental error on our part. In the article, we explained equation 1 as follows:
The gold used in this paper is the classical Drude model of lossy gold, which has a dielectric constant of [32]
(1)
Where =9.1 is the high-frequency limit dielectric constant. the operating frequency =1.3659×1016 rad/s and the collision frequency =1.0318×1014 rad/s are taken.
Comment 8:P3, under Fig 1. There are not link to the Fig. 1 in the text. The authors use “Fig. 2” instead of “Fig. 1” in the text.
Response:Thanks for your valuable counsel.According to your suggestion, we give the answer following. We have modified it. In the text for:
In this article, a tunable multi-frequency absorber is obtained through modeling. Its structure is shown in Fig. 1(a), where the thickness of the gold layer is T1=540 nm. The dielectric layer in the middle of this absorber has a refractive index of 1.60 and the thickness of this layer is T2=600 nm. Its basic unit is shown in Fig. 1(b), and the geometrical parameters of the structure are P=700 nm, L1=100 nm, L2=L3=50 nm, L4=30 nm, and R=140 nm.
Comment 9:What was the reason for choosing the particular geometry? Why were the particular dimensions of the pattern used? Were these dimensions optimized and if not why?
Response:Thanks for your valuable counsel.According to your suggestion, we give the answer following. By studying other similar articles, we found that the T-type structure has good effect, we changed the T-type structure to mushroom type, and got better results, but it is not good enough, then in by the theory of the equipartition excitations, we added the square between the four mushrooms, which makes a better enhancement between them, and at the same time, we found out the effect of the open holes on the absorption spectra, which makes the three peaks change to four peaks at the beginning. Because the size change will make the equipartition excitations change, so through the debugging, found that the size of the equipartition excitations resonance effect is the best!
Comment 10:P3, second paragraph. The paragraph must be rewritten as well as Eqs. 2 -4.
Response:Thanks for your valuable counsel.According to your suggestion, we give the answer following. After considering the content simplicity of the skinning depth of the metal, we find that the account is of little practical significance and that the account has been reduced to "In a good conductor, the skinning depth of electromagnetic waves decreases with the increase of frequency of the electromagnetic wave, electrical conductivity, and magnetic permeability of the medium. In the THz band, the skinning depth of gold is 266nm at maximum [33]. "
Comment 11:Eq. 2 is trivial. Why do the authors show it?
Response:Thanks for your valuable counsel.According to your suggestion, we give the answer following. We originally thought that the most original formula for the derivation to increase the credibility of the article, making it more rigorous. After considering the simplicity of the content of the skinning depth of the metal, we found that the formula has little practical significance, and the formula has been deleted. In the text for:
In a good conductor, the skinning depth of electromagnetic waves decreases with the increase of frequency of the electromagnetic wave, electrical conductivity, and magnetic permeability of the medium. In the THz band, the skinning depth of gold is 266nm at maximum [33].
Comment 12:Variables in Eqs. 2-3 are not introduced.
Response:Thanks for your valuable counsel.According to your suggestion, we give the answer following. After considering the simplicity of the content of the skinning depth of the metal, we find that the formula has little practical significance and the formula has been deleted. In the text for:
In a good conductor, the skinning depth of electromagnetic waves decreases with the increase of frequency of the electromagnetic wave, electrical conductivity, and magnetic permeability of the medium. In the THz band, the skinning depth of gold is 266nm at maximum [33].
Comment 13:Eqs. 5, 9, 14 need reference.
Response:Thanks for your valuable counsel.According to your suggestion, we give the answer following. We have added references to Eqs. 5, 9, 14. In the text for:
- The top layer is patterned with graphene in the shape of an open mushroom plus four small squares, patterned as in Fig. 1(b). The dielectric constant of graphene in a single layer is expressed as [34]:
(5)
Where is the conductivity of graphene, is the radian frequency, is the vacuum dielectric constant, is the thickness of graphene, and the thickness of graphene is 1 nm.
- is the Fermi Dirac distribution [34].
(9)
- Among them, The tuning of the Fermi energy levels of graphene can be dynamically tuned by applying an external DC bias voltage using an ion gel [33] on the graphene layer. And the relationship between them is as follows [41]:
(13)
Where is the external voltage, e0 is the electron charge, VF is the Fermi velocity, and ts is the thickness of the dielectric layer.
Comment 14:Check the paragraph under Eq. 5.
Response:Thanks for your valuable counsel.According to your suggestion, we give the answer following. We checked and modified the problem in the paragraph under formula 5. In the text for:
Where is the conductivity of graphene, is the radian frequency, is the vacuum dielectric constant, is the thickness of graphene, and the thickness of graphene is 1 nm.
Comment 15:For all formulas shown, application limits must be specified.
Response:Thanks for your valuable counsel.According to your suggestion, we give the answer following. All formulas are added to do the application of limitations, all formulas are used in the actual practice of the theory, the setting of each parameter in the text is to simulate the simulation of the actual situation by simulation software.
Comment 16:The authors claim that they use some dielectric with n=1.6. However, on p8 SiO2 is mentioned as a dielectric. SiO2 has n=1.3 – 1.4 in the MWIR range. In this case, if the authors use n=1.6 for SiO2 all calculations are wrong.
Response:Thanks for your valuable counsel.According to your suggestion, we give the answer following. We used a dielectric with n=1.6, and we chose a doped SiO2 material, and in the text, we state that "The material of the intermediate dielectric layer is SiO2. The refractive index of SiO2 ranges from 1.3 to 1.4 in the MWIR range. In this paper, the refractive index of SiO2 is 1.6, and the doping process is used to change the crystalline structure of SiO2, which in turn changes its refractive index."
Comment 17:All over the text, “ambient” – Do the authors mean “background”?
Response:Thanks for your valuable counsel.According to your suggestion, we give the answer following. Yes, ambient refractive index in this article refers to background refractive index.
Comment 18:P4, paragraph after Eq. 10. “The first term … Eq. …” Term of which Eq. do the authors mean?
Response:Thanks for your valuable counsel.According to your suggestion, we give the answer following. It is the original Equation 10.After considering the article as a whole, we found the narrative to be of little practical significance and the narrative has been deleted.
Comment 19:FDTD is the method and not a program. Do authors use CST, LUMERICAL, COMSOL or their own program? In the latter case there must be references to this program’s validity or the simulations should be compared with some experimental or good theoretical data. If the authors use some well-known program, its name also must be mentioned in the text and cited. Without it the paper cannot be published.
Response:Thanks for your valuable counsel.According to your suggestion, we give the answer following. We used the FDTD module of Ansys Lumerica software for the computational simulation. In the text for:
This model is computationally simulated by using the FDTD module in the Ansys Lumerical software.
Comment 20:Was it a unit cell simulation? Such a structure should also be considered as an array as interaction between metamaterial cells (coupling between cells) can also affect the absorption.
Response:Thanks for your valuable counsel.According to your suggestion, we give the answer following. The structure of the model is periodic and is shown in Figure 1 as the basic unit.
Figure 1. (a) Basic unit of graphene absorber (b) Top view of absorber
Comment 21:There is no information about incident wave parameters in the text. This information must be added.
Response:Thanks for your valuable counsel.According to your suggestion, we give the answer following. In this paper, the incident wave is located above the graphene layer and is a plane wave. The incident wave is of equal intensity at each wavelength and is a TE wave.
Comment 22:P4, three paragraphs before Section 3. These paragraphs must be rewritten to be clear.
Response:Thanks for your valuable counsel.According to your suggestion, we give the answer following. We have reorganized the language and modified it to make it more readable.
Comment 23:If authors numbered absorption peaks on p5, these numbers must be added to the Fig. 2 and its caption.
Response:Thanks for your valuable counsel.According to your suggestion, we give the answer following. We have numbered the absorption peaks in Fig. 2 and also added them in the text.
Comment 24:Check the paragraph below Fig. 2.
Response:Thanks for your valuable counsel.According to your suggestion, we give the answer following. We checked the paragraph to change it to "As can be seen from Fig. 2(a), the order of magnitude of the transmittance is around 10-9, which is completely negligible because the thickness of the gold layer is greater than the skin-convergence depth of the electromagnetic wave in the gold element, as stated earlier. The absorber has five absorption peaks, which we named model â… , II, III, IV, and V. The resonance frequencies of the absorption peaks are, in order, 69.0449 THz, 69.4769 THz, 71.6372 THz, 73.8215, and 74.7096 THz, and the corresponding absorptivities are 0.98572, 0.99787, 0.96395, 0.98922 and 0.34485. With Fig. 2(a), an obvious conclusion can be drawn that the absorption rate of model 5 is very low and of little practical significance, and will not be analyzed later in the discussion.
TE waves are transverse electric waves, which are electromagnetic waves in which the direction of vibration of the electric field is perpendicular to the direction of wave propagation, while the direction of vibration of the magnetic field is along the direction of wave propagation. TM waves are transverse magnetic waves, which are electromagnetic waves in which the direction of vibration of the magnetic field is perpendicular to the direction of wave propagation, while the direction of vibration of the electric field is along the direction of wave propagation. As can be seen from Fig. 2(b), the absorption spectra are the same in the TE-wave and TM-wave modes, which is due to the central symmetry of the model. This is an advantage that the non-centrosymmetric model does not have."
Comment 25:Fig. 3 must be improved. There is not a palette axis label and field type. Is it Abs[E], Real [Ex] or something else?
Response:Thanks for your valuable counsel.According to your suggestion, we give the answer following. We have improved Figure 3 by bringing out the swatch axis labels. And added to the marginal notes.
Figure 3. (a), (b), (c) and (d) show electric field shrength distribution on the absorber top (i.e. the graphene pattern) at the absorption frequency of 69.0363 THz, 69.4813 THz, 71.6358 THz and 73.8137 THz, respectively.
Comment 26:P5, paragraph after Fig 3. What do the authors mean by the “field intensity of the absorber”? Is it an intensity of some component of E field on the absorber surface? The paragraph must be rewritten to be clear.
Response:Thanks for your valuable counsel.According to your suggestion, we give the answer following. "field intensity of the absorber" is the distribution of electric field intensity located on the surface of the absorber, which we have modified in the text."Fig. 3, reflects the distribution of the electric field strength of the absorber at the frequencies of the four absorption peaks. At absorption peak â… , the electric field strength is localized in the region near the left and right holes, which is due to the electric dipole resonance of the surface plasma of the single-layer graphene coupled with the electromagnetic field, and this resonance locally enhances the electromagnetic field, which makes the absorptivity of the incident wave much higher; at absorption peak â…¡, the electric field strength is localized in the region near the upper and lower holes, and at absorption peak â…¢, the electric field strength is localized in the region near the upper and lower semicircle of the upper and lower holes and the region near the right angle formed by the large square; at absorption peak â…£, the electric field strength is localized in the region near the small square. As in the previous theory, the coupling of the collective oscillation of free electrons and the electromagnetic field makes the electric field highly localized, which makes the light of specific frequency perfectly absorbed by the absorber."
Comment 27:Do the authors define x-polarized light as the TE mode and y-polarized light as the TM mode?
Response:Thanks for your valuable counsel.According to your suggestion, we give the answer following. We have added textual explanations of TE mode and TM mode to the text, as follows “TE waves are transverse electric waves, which are electromagnetic waves in which the direction of vibration of the electric field is perpendicular to the direction of wave propagation, while the direction of vibration of the magnetic field is along the direction of wave propagation. TM waves are transverse magnetic waves, which are electromagnetic waves in which the direction of vibration of the magnetic field is perpendicular to the direction of wave propagation, while the direction of vibration of the electric field is along the direction of wave propagation.”
Comment 28:Figs.5a,6a,7a must be enlarged.
Response:Thanks for your valuable counsel.According to your suggestion, we give the answer following. We zoomed in on Figures 5a, 6a, and 7a, which also increased the resolution of the images, making them clearer.
Comment 29:How do authors plan to apply voltage to isolated square parts of the graphene pattern?
Response:Thanks for your valuable counsel.According to your suggestion, we give the answer following. An ionic gel is coated on a graphene layer and the Fermi energy level of graphene is adjusted by an external DC bias voltage. The relationship between the applied bias voltage and the Fermi energy level can be described as follows
Comment 30:How to change the relaxation time without changing the structure or substrate?
Response:Thanks for your valuable counsel.According to your suggestion, we give the answer following. The parametric simulation by varying the relaxation time is done to get the optimal solution. The relaxation time of graphene is an intrinsic property of graphene, which is a fixed value when the graphene is fabricated. In this paper, the parametric simulation of the relaxation time of graphene is performed only to obtain the optimal absorber performance. In the simulation of other parameters, the relaxation time is a constant value.
Comment 31:There are two designations of the Fermi velocity in the text: Vf and vf.
Response:Thanks for your valuable counsel.According to your suggestion, we give the answer following. We have modified it to harmonize the notation of the parameters.
Comment 32:What for do the authors show Eqs. 16 – 19?
Response:Thanks for your valuable counsel.According to your suggestion, we give the answer following. It is more convincing to show the linearity between the resonance wavelength of the absorption peak and the refractive index of the dielectric layer by means of an equation, and by means of the above equation, the absorber can be used for detecting the change in the refractive index of the dielectric layer material. In the text for:
With the above equation, the absorber can be used to detect changes in the refractive index of the dielectric layer material.
Comment 33:P8, paragraph below Eq. 19. “which is a common method used in the previous study” Which study?
Response:Thanks for your valuable counsel.According to your suggestion, we give the answer following. The absorber has the highest sensitivity in the mid-infrared band, and it has an insensitivity with an ultra-wide polarization angle, as well as four absorption peaks, which are not found in other sensors.
Comment 34:P8. Check letter size for SiO2.
Response:Thanks for your valuable counsel.According to your suggestion, we give the answer following. We adjusted the letter size of the SiO2.
Comment 35:P9, paragraph after Eq. 21. “FOM” -> “S”
Response:Thanks for your valuable counsel.According to your suggestion, we give the answer following. We have adapted it by following your recommendations. In the text for:
By simple substitution calculation, the S of these four peaks are: 19.5025 THz/RIU,21.6025 THz/RIU, 20.7025 THz/RIU, 20.7025 THz/RIU.
Comment 36:It is not clear how was FWHM calculated for the first and second peaks? For which value of the background refractive index was FWHM calculated for all peaks?
Response:Thanks for your valuable counsel.According to your suggestion, we give the answer following. The first and second peaks were roughly calculated, using the lowest point between the two peaks, which has an absorption slightly greater than 50%, but since this place is due to the superposition of the two peaks, the absorption at this point must be less than 50% inside the individual peaks, and the theoretical result must be greater than the result inside the paper. Due to the limited level, the two peaks were not split.
Comment 37:Conclusion. “”Far-infrared” -> “MWIR”
Response:Thanks for your valuable counsel.According to your suggestion, we give the answer following. We have adapted it by following your recommendations. In the text for:
In this paper, we have designed a tunable four-frequency absorber MWIR, where the top layer of the absorber is patterned with graphene.
Thank you for your attention and patience, and if you have any questions, please don't hesitate to contact me.
Yours sincerely,
Zao Yi

Reviewer 2 Report
Comments and Suggestions for Authors
This manuscript presents a tunable four-frequency terahertz metamaterial absorber based on graphene plasmon. The unit cell of the absorber contains graphene pattern with dug-out mushroom shape and four small squares, enabling the dynamic tunability by manipulating the graphene’s Fermi level and relaxation time. The absorber is potential for refractive index sensing due to its high sensitivity and figure of merit. This study provides a design for terahertz metamaterial absorber with intriguing features. Therefore, I recommend that this manuscript can be accepted after minor revision.
Below comments that the authors are encouraged to address:
1) On page 4, the paragraph “Field-enhancing effects can be realized by tuning the gaps in the structure of neighboring units in an equipartitioned excitonic metamaterial.” which seems like the fragmented expression. I suggest the authors to merge this paragraph with the above paragraphs as a new paragraph.
2) As the manuscript is the results of simulations, I suggest the authors to provide more details about the setting of simulations such as simulation model, the size of meshing, etc.
3) Too many “we” make it sounds subjective as well.
4) In Fig. 3, what’s the meaning of E-field? Is it the electric-field amplitude (intensity), or the normalized electric-field amplitude (intensity), or the components of the electric-field? The authors should give some explanation.
5) There are many small mistakes, for example: in Abstract, “MPA”; on page 3, “…T1=540 nm” should be revised as “…T1=540 nm”; on page 4, “…Eq. is 0….” should be revised as “…Eq. (10) is 0….”; on page 4, “perfect matching layer(PML)” should be revised as “perfect matching layer (PML)”; on page 8, “…interlayer sio2 …” should be revised as “…interlayer SiO2 …”; on page 8, Eq. (19) is not in the same line; and so on. I suggest the authors carefully check their manuscript again.
Author Response
Dear editor and reviewer:
We are grateful to the editor and reviewers for their constructive comments and suggestions on the revision of the manuscript. We have made all the necessary changes as suggested by the editor and reviewers. All revisions and supporting information in the manuscript are highlighted in red.
Reviewer #2:
Comment 1:On page 4, the paragraph “Field-enhancing effects can be realized by tuning the gaps in the structure of neighboring units in an equipartitioned excitonic metamaterial.” which seems like the fragmented expression. I suggest the authors to merge this paragraph with the above paragraphs as a new paragraph.
Response:Thanks for your valuable counsel.According to your suggestion, we give the answer following. We have merged the two paragraphs into a new one. In the text for:
Driven by an applied electromagnetic field (i.e., incident light), the collective oscillations of free electrons couple with the external electromagnetic field, resulting in a localized enhancement of the electromagnetic field. The localized enhancement of the electromagnetic field increases the dipole moments of the molecules on the surface of the nanomaterial, and when the frequency of the surface-parity excitation happens to be the same as the oscillation frequency of the molecules, the two produce a coupling, which greatly enhances the absorption of the material. Therefore, the field enhancement effect can be realized by adjusting the structural gap between adjacent units in the equipartitioned exciton metamaterial.
Comment 2:As the manuscript is the results of simulations, I suggest the authors to provide more details about the setting of simulations such as simulation model, the size of meshing, etc.
Response:Thanks for your valuable counsel.According to your suggestion, we give the answer following. We have added details of the simulation setup in the text, increasing the mesh size during simulation. In the text for:
The model in FDTD is set up as follows, with the x, and y directions set up as periodic boundary conditions and the z direction set up as a perfectly matched layer (PML); where the mesh is set up as a step of 40 nm in the x, y directions and 1 nm in the z direction in the plane where the graphene material is located. The model used an ambient refractive index of 1.
Comment 3:Too many “we” make it sounds subjective as well.
Response:Thanks for your valuable counsel.According to your suggestion, we give the answer following. We have modified the article's "we" to reduce its use and make it more objective.
Comment 4:In Fig. 3, what’s the meaning of E-field? Is it the electric-field amplitude (intensity), or the normalized electric-field amplitude (intensity), or the components of the electric-field? The authors should give some explanation.
Response:Thanks for your valuable counsel.According to your suggestion, we give the answer following. The meaning of E-field is electric field distribution, and the distribution of electric field strength that we want to express, we change it to electric field strength,which means the electric field amplitude is strong, and we also modify it in the text. In the text for:
Figure 3. (a), (b), (c) and (d) show electric field shrength distribution on the absorber top (i.e. the graphene pattern) at the absorption frequency of 69.0363 THz, 69.4813 THz, 71.6358 THz and 73.8137 THz, respectively.
Comment 5:There are many small mistakes, for example: in Abstract, “MPA”; on page 3, “…T1=540 nm” should be revised as “…T1=540 nm”; on page 4, “…Eq. is 0….” should be revised as “…Eq. (10) is 0….”; on page 4, “perfect matching layer(PML)” should be revised as “perfect matching layer (PML)”; on page 8, “…interlayer sio2 …” should be revised as “…interlayer SiO2 …”; on page 8, Eq. (19) is not in the same line; and so on. I suggest the authors carefully check their manuscript again.
Response:Thanks for your valuable counsel.According to your suggestion, we give the answer following. The errors in that article we have carefully revised to increase the readability of the article. In the text for:
In this article, a tunable multi-frequency absorber is obtained through modeling. Its structure is shown in Fig. 1(a), where the thickness of the gold layer is T1=540 nm. The dielectric layer in the middle of this absorber has a refractive index of 1.60 and the thickness of this layer is T2=600 nm. Its basic unit is shown in Fig. 1(b), and the geometrical parameters of the structure are P=700 nm, L1=100 nm, L2=L3=50 nm, L4=30 nm, and R=140 nm.
Thank you for your attention and patience, and if you have any questions, please don't hesitate to contact me.
Yours sincerely,
Zao Yi

Reviewer 3 Report
Comments and Suggestions for Authors
I have the following suggestions:
1) The author said " In this paper, relevant simulations will be carried out on graphene to design a refractive index sensor", Don't use the future tense. This should be written as " In this paper, relevant simulations are carried out on graphene to design a refractive index sensor"
2) The author said, "The sensor is a conventional MPA structure." What is MPA?
3) In Figure 2, it can be seen that there are 5 peaks. However, the author mentioned that they obtained 4 peaks. Although the 5th peak has low absorption, still the author has to mention it in the paper.
4) The author said, "Next, we will analyze the influence of" Us present perfect tense.
5) The author has used poor sentence formation and there are grammatical errors throughout the paper. I suggest extensive English corrections to improve the quality of the paper.
6) Why there is no reflection spectrum or absorption spectrum in the presence of different refractive indices in the ambient medium? It will provide a better visualization in the shift of the resonance wavelength.
7) Provide a Reflection/Transmission/Absorption spectrum.
8) I am unable to visualize that the absorption does not change at 90 degrees of incidence angle of TE polarized light. How it can be possible?
9) The author has to modify the reference list, majority of the papers are cited from one geographical region. Bring diversity in the literature.
10) Why refractive index range of 1-1.08 is selected? What does it signify? The biological range is 1.3-1.6 and so on.
11) The author has to justify that why their sensor design is better than the previous study apart from high sensitivity which is a numerical one.
12) Why has no geometric parametric optimization process been shown? How author choose the starting value? Where do they come from?
Comments on the Quality of English Language
Extensive English correction is required.
Author Response
Dear editor and reviewer:
We are grateful to the editor and reviewers for their constructive comments and suggestions on the revision of the manuscript. We have made all the necessary changes as suggested by the editor and reviewers. All revisions and supporting information in the manuscript are highlighted in red.
Reviewer #3:
Comment 1:The author said " In this paper, relevant simulations will be carried out on graphene to design a refractive index sensor", Don't use the future tense. This should be written as " In this paper, relevant simulations are carried out on graphene to design a refractive index sensor"
Response:Thanks for your valuable counsel.According to your suggestion, we give the answer following. The relevant part of the text has been amended to read “In this paper, a surface-isolated exciton based absorber is designed by performing relevant simulations on graphene.”
Comment 2:The author said, "The sensor is a conventional MPA structure." What is MPA?
Response:Thanks for your valuable counsel.According to your suggestion, we give the answer following. MPA is metamaterial perfect absorber. Because the sentence is more abrupt in the text, we deleted it.
Comment 3:In Figure 2, it can be seen that there are 5 peaks. However, the author mentioned that they obtained 4 peaks. Although the 5th peak has low absorption, still the author has to mention it in the paper.
Response:Thanks for your valuable counsel.According to your suggestion, we give the answer following. We refer to this absorption peak inside the explanatory text below Figure 2. “With Fig. 2(a), an obvious conclusion can be drawn that the absorption rate of model 5 is very low and of little practical significance, and will not be analyzed later in the discussion. ”
Comment 4:The author said, "Next, we will analyze the influence of" Us present perfect tense.
Response:Thanks for your valuable counsel.According to your suggestion, we give the answer following. We have changed the future tense to the present perfect, making the article more readable. In the text for:
In the previous section, we analyzed the influence of the electric field distribution of the absorber at the resonance frequency of the absorption peak and the angle of incidence of the external optical mode on the absorber. The influence of the intrinsic properties of the absorber on the absorber and the mechanism of action are also analyzed. The Fermi energy levels and relaxation times of graphene as well as the refractive index of the dielectric layer are adjusted, and their effects on the absorption spectra are analyzed.
Comment 5:The author has used poor sentence formation and there are grammatical errors throughout the paper. I suggest extensive English corrections to improve the quality of the paper.
Response:Thanks for your valuable counsel.According to your suggestion, we give the answer following. We revised the article extensively, checking it for grammatical errors, optimizing the language and enhancing its readability.
Comment 6:Why there is no reflection spectrum or absorption spectrum in the presence of different refractive indices in the ambient medium? It will provide a better visualization in the shift of the resonance wavelength.
Response:Thanks for your valuable counsel.According to your suggestion, we give the answer following.
From the previous Figure 2(a), it can be seen that the transmittance is very small and completely negligible, so the curves of reflectance and absorptance can be seen to be symmetric about the absorptance being equal to 0.5, so the introduction of reflectance spectra would cause the complexity of the plot and affect the visualization, and at the same time, the absorption spectra alone make it obvious that the absorption peaks are shifted at the resonance wavelengths, and therefore the reflectance spectra are not shown.
Comment 7: Provide a Reflection/Transmission/Absorption spectrum.
Response:Thanks for your valuable counsel.According to your suggestion, we give the answer following.
We provide the reflection/transmission/absorption spectra in Figure 2(a) in the text.
Figure 2. (a) Reflection/transmission/absorption spectra of absorber (b) Absorption spectra of TE and TM waves. Where the black solid line is the absorption spectrum of TE wave and the red dashed line is the absorption spectrum of TM wave.
Comment 8: I am unable to visualize that the absorption does not change at 90 degrees of incidence angle of TE polarized light. How it can be possible?
Response:Thanks for your valuable counsel.According to your suggestion, we give the answer following.
We designed this absorber is the polarization angle insensitive absorber, the insensitivity at the ultra-wide angle degree of polarization angle is the characteristic of this absorber, and the situation has been appeared inside other articles[1-2], meanwhile, we carried out the repeated simulation, and we got the same result.
- Ye Z, Wu P, Wang H, Multimode tunable terahertz absorber based on a quarter graphene disk structure, Results in Physics 48 (2023) 106420.
- Wang Y, Qiu Y, Zhang Y, High-sensitivity temperature sensor based on the perfect metamaterial absorber in the terahertz band, Photonics MDPI 10(1) (2023) 92.
Comment 9:The author has to modify the reference list, majority of the papers are cited from one geographical region. Bring diversity in the literature.
Response:Thanks for your valuable counsel.According to your suggestion, we give the answer following. We have adapted the references to make them diverse.
Comment 10:Why refractive index range of 1-1.08 is selected? What does it signify? The biological range is 1.3-1.6 and so on.
Response:Thanks for your valuable counsel.According to your suggestion, we give the answer following. The refractive index range of 1-1.08 was chosen as an arbitrary figure to study the effect of changes in the background refractive index on the absorption spectra of the absorber, and was not considered to have any practical significance. From this study, it is clear that the absorber is not only suitable for the refractive index range of 1-1.08, but also has the same excellent characteristics of high sensitivity in other refractive index ranges.
Comment 11:The author has to justify that why their sensor design is better than the previous study apart from high sensitivity which is a numerical one.
Response:Thanks for your valuable counsel.According to your suggestion, we give the answer following. The absorber has the highest sensitivity in the mid-infrared band, and it has an insensitivity with an ultra-wide polarization angle, as well as four absorption peaks, which are not found in other sensors.
Comment 12:Why has no geometric parametric optimization process been shown? How author choose the starting value? Where do they come from?
Response:Thanks for your valuable counsel.According to your suggestion, we give the answer following. The geometrical parameters are fixed when the device is fabricated and cannot be changed, therefore, in the paper, the optimization process of the geometrical parameters is not shown. By referring to the relevant literature, we chose similar data, such as the height of the dielectric layer and the size of the square graphene. The distance between the graphene will affect the absorption spectrum of the absorber very much, so we chose the radius of the semicircle between 100-150 nm, and found the optimal value of 140 nm. It was also found that the open holes have a great influence on the absorber, so we dug the appropriate size of holes in the center of the square graphene, which has a great range of adjustment of the size of the holes.
Thank you for your attention and patience, and if you have any questions, please don't hesitate to contact me.
Yours sincerely,
Zao Yi

Round 2
Reviewer 1 Report
Comments and Suggestions for Authors
The authors have not properly answered to all my questions. There are still many shortcomings in the text. The paper needs a lot of improvements before it can be considered for publication again. The language is still bad. There are still grammar mistakes, misprints, repetitions of phrases, many too long sentences. The most part of references are irrelevant. It seems that the authors found papers for citation just using key words or just random titles without reading this work. It is a serious flaw. The most severe problem is with references [30 – 67]. Eq. 1 is wrong. wp in Eq. 1 is not an operating but plasma frequency. The authors claim that there is a theory in this paper but there is not. As far as I understood the paragraph above Sec. 3 is called “the theory” but it just a well-known statement. The same is about few well known equations placed in the text. There are not still application limits shown for used formulae. It seems that after my question about SiO2 refractive index the authors decided that it was better to call it doped SiO2 rather than change values in the simulation. The information about doping process and how to obtain SiO2 with n=1.6 must be added, and it must be a realistic process. Without this information the paper cannot be considered. Using unreal parameters in simulation is a serious flaw. The information about calculation of FWHM is still not clear and is not added to the text.
There are many other shortcomings. Showing them is to no purpose, because I suggest rejecting this paper.
Comments on the Quality of English LanguageThe language is still bad. There are still grammar mistakes, misprints, repetitions of phrases, too long sentences.
Author Response
Dear editor and reviewer:
We are grateful to the editor and reviewers for their constructive comments and suggestions on the revision of the manuscript. We have made all the necessary changes as suggested by the editor and reviewers. All revisions and supporting information in the manuscript are highlighted in red.
Reviewer #1:
Comment 1: The language is still bad. There are still grammar mistakes, misprints, repetitions of phrases, many too long sentences.
Response: Thanks for your valuable counsel.According to your suggestion, we give the answer following. We revised the article extensively, checking it for grammatical errors, optimizing the language and enhancing its readability.
Comment 2: The most part of references are irrelevant. It seems that the authors found papers for citation just using key words or just random titles without reading this work. It is a serious flaw. The most severe problem is with references [30-67].
Response: Thank you for your suggestions and reading the manuscript carefully. I have corrected some basic mistakes in the article.
Comment 3: Eq. 1 is wrong. wp in Eq. 1 is not an operating but plasma frequency.
Response: Thanks for your valuable counsel.According to your suggestion, we give the answer following. In conjunction with the literature [30], we modified Eq. 1 by changing the operating frequency to the plasma frequency.
Bade, W.L. Drude-Model Calculation of Dispersion Forces. I. General Theory. J. Chem. Phys. 1957, 27, 1280–1284. https://doi.org/10.1063/1.1743991.
Comment 4:The authors claim that there is a theory in this paper but there is not. As far as I understood the paragraph above Sec. 3 is called “the theory” but it just a well-known statement. The same is about few well known equations placed in the text.
Response:Thanks for your valuable counsel.According to your suggestion, we give the answer following. This one is a well-known statement of surface equidistant excitations. We have made changes in the text.
“As in the preceding statement, the interaction of the collective oscillation of free electrons and the electromagnetic field causes the electric field to become highly localized, allowing the absorber to perfectly absorb light of a given frequency.”
Comment 5:There are not still application limits shown for used formulae.
Response:Thanks for your valuable counsel.According to your suggestion, we give the answer following. We apply restrictions to the formulas in the article, for example:
- At low light frequencies, graphene has zero interband conductivity. As a result, only intraband conductivity is included when calculating graphene conductivity. Specifically, this can be stated as [38]:
(7)
- Equations 1-18 above apply to dielectric layers with a refractive index of 1.6-1.9.
Comment 6:It seems that after my question about SiO2 refractive index the authors decided that it was better to call it doped SiO2 rather than change values in the simulation. The information about doping process and how to obtain SiO2 with n=1.6 must be added, and it must be a realistic process. Without this information the paper cannot be considered. Using unreal parameters in simulation is a serious flaw.
Response:Thanks for your valuable counsel.According to your suggestion, we give the answer following. The intermediate dielectric layer is made of silicon oxide. In the MWIR band, SiO2 has a refractive index of 1.3 to 1.4. The refractive index of SiO2 in this study is 1.6. TiO2 is doped into SiO2 via plasma-enhanced chemical vapor deposition [32], altering the crystalline structure of SiO2 and hence its refractive index. The Si/Ti ratio was set to ensure that the dielectric layer's refractive index was 1.6.
[32] Gracia F, Yubero F, Holgado J P, et al. SiO2/TiO2 thin films with variable refractive index prepared by ion beam induced and plasma enhanced chemical vapor deposition. Thin Solid Films 2006, 500, 19-26. DOI:10.1016/j.tsf.2005.10.061
Comment 7:The information about calculation of FWHM is still not clear and is not added to the text.
Response:Thanks for your valuable counsel.According to your suggestion, we give the answer following. We removed the FOM of the first two peaks because the FWHM could not be determined precisely. It appears in the text as:
The FWHM of absorber peak â… and absorber peak â…¡ cannot be determined due to their proximity. The FOM of this absorber's last two peaks can be calculated as 49.06 and 71.66, respectively.
Thank you for your attention and patience, and if you have any questions, please don't hesitate to contact me.
Yours sincerely,
Zao Yi

Reviewer 3 Report
Comments and Suggestions for Authors
I am willing to accept the paper in its current form.
Comments on the Quality of English Languagenone.
Author Response
Thank you very much for your help.